# Clinical Characteristics, Treatments and Outcomes of 18 Lung Transplant Recipients with COVID-19

René Hage [1,2,*], Carolin Steinack [1,2], Fiorenza Gautschi [1,2], Susan Pfister [3], Ilhan Inci [4] and Macé M. Schuurmans [1,2]

1   Division of Pulmonology, University Hospital Zurich, CH 8091 Zurich, Switzerland; carolin.steinack@usz.ch (C.S.); fiorenza.gautschi@usz.ch (F.G.); mace.schuurmans@usz.ch (M.M.S.)
2   Faculty of Medicine, University of Zurich, CH 8091 Zurich, Switzerland
3   Division of Internal Medicine, University Hospital Zurich, CH 8091 Zurich, Switzerland; susan.pfister@usz.ch
4   Department of Thoracic Surgery, University Hospital Zurich, CH 8091 Zurich, Switzerland; Ilhan.inci@usz.ch
*   Correspondence: rene.hage@usz.ch; Tel.: +41-44-2559111

**Abstract:** We report clinical features, treatments and outcomes in 18 lung transplant recipients with laboratory confirmed SARS-CoV-2 infection. We performed a single center, retrospective case series study of lung transplant recipients, who tested positive for SARS-CoV-2 between 1 February 2020 and 1 March 2021. Clinical, laboratory and radiology findingswere obtained. Treatment regimens and patient outcome data were obtained by reviewing the electronic medical record. Mean age was 49.9 (22–68) years, and twelve (67%) patients were male. The most common symptoms were fever (n = 9, 50%), nausea/vomiting (n = 7, 39%), cough (n = 6, 33%), dyspnea (n = 6, 33%) and fatigue (n = 6, 33%). Headache was reported by five patients (28%). The most notable laboratory findings were elevated levels of C-reactive protein (CRP) and lactate dehydrogenase (LDH). Computed Tomography (CT) of the chest was performed in all hospitalized patients (n = 11, 7%), and showed ground-glass opacities (GGO) in 11 patients (100%), of whom nine (82%) had GGO combined with pulmonary consolidations. Six (33%) patients received remdesivir, five (28%) intravenous dexamethasone either alone or in combination with remdesivir, and 15 (83%) were treated with broad spectrum antibiotics including co-amoxicillin, tazobactam-piperacillin and meropenem. Four (22%) patients were transferred to the intensive care unit, two patients (11%) required invasive mechanical ventilation who could not be successfully extubated and died. Eighty-nine percent of our patients survived COVID-19 and were cured. Two patients with severe COVID-19 did not survive.

**Keywords:** SARS-CoV-2; viral infection; hyperinflammation; cytokine storm syndrome; dexamethasone; remdesivir; hemodialysis

## 1. Introduction

In the last decades, life expectancy in lung transplant recipients (LTRs) has improved. According to data of the International Society of Heart and Lung Transplant registry, including 260 lung transplantation centers with 69,200 adult LTR worldwide, infections remain the leading cause of death within the first year after lung transplantation [1]. Coronavirus disease 2019 (COVID-19), caused by the severe acute respiratory syndrome coronavirus 2 (SARS CoV-2) has affected solid organ transplant (SOT) recipients as well as immunocompetent patients and continues to claim lives globally. Data on COVID-19 in LTRs are scarce, mostly reported in case reports and small case series. One study reported 17 LTR with COVID-19 in a series of 90 SOT recipients, but the LTR were not analyzed separately [2]. Verleden et al. studied a total of 10 LTR with COVID-19 [3]. Another study performed a telephone survey including 41 LTR with COVID-19 [4].

Surprisingly, with respect to the chronic immunosuppression in SOT recipients, the expected higher incidence and mortality of COVID-19 in this group has not been widely observed. One hypothesis is that the use of calcineurin inhibitors in this group of patients



mitigates the severe hyperinflammation (cytokine storm syndrome) and may thus contribute to less morbidity and mortality. This retrospective study of consecutive LTRs with COVID-19 from the Zurich Lung Transplant Center aims to describe the features of the disease by analyzing the clinical, laboratory and radiology characteristics and the outcome of these patients.

## 2. Methods

### 2.1. Study Population, Setting, and Clinical Data

The study population consists of consecutive adult LTR recipients ≥18 year of age with COVID-19 at the University Hospital Zurich, Switzerland, in which formal informed consent was given. No patient had to be excluded due to lack of informed consent or age <18 years. For the diagnosis of COVID-19, a laboratory confirmation of SARS CoV-2 by real time reverse transcriptase polymerase chain reaction (RT-PCR) was required, irrespective of clinical signs and symptoms. The data comprised demographics, clinical, laboratory and radiology characteristics as well as treatments and outcomes. The data were abstracted from the electronic medical record of the patients. The long-term immunosuppression was recorded. We also documented comorbidity including hypertension, diabetes, cardiovascular disease, malignancy and chronic kidney disease.

Severity of disease was classified according to Siddiqi [5]. The risk stratification by the AIFELL score was documented [6]. Data on antiviral, antibiotic and anti-inflammatory therapy were recorded, as well as clinical treatment setting (normal ward, intermediate care, and intensive care). Additionally, we recorded the need for oxygen therapy (normal breathing without additional oxygen, oxygen therapy with nasal cannula, non-rebreather face mask, high flow oxygen therapy, non-invasive ventilation, invasive mechanical ventilation or extracorporeal membrane oxygenation). Full recovery was defined as two negative SARS CoV-2 RT-PCR tests at least 24 h apart along with resolution of symptoms and clinical syndrome, and in case of hospital discharge without the need for additional oxygen therapy.

### 2.2. Laboratory and Radiology Assessment

Diagnosis of COVID-19 was confirmed by RT-PCR using nasopharyngeal swabs. Laboratory investigations included complete blood count (hemoglobin, leucocytes and platelets) with differential blood count including eosinophils, neutrophils, and lymphocytes. The chemistry panel included the renal function, liver enzymes, and the C-reactive protein (CRP). Radiology data included computed tomography (CT) scan of chest in all hospitalized patients.

### 2.3. Statistical Analysis

Descriptive statistics were performed. Main data are summarized in tables. Results were reported as mean with the range, and categorical variables were calculated as counts (n) and percentages (%).

### 2.4. Ethical Consideration

The study was granted approval by the Zurich branch of the Swiss Medical Ethics Committee (Swissethics, No. 2021-00293).

## 3. Results

### 3.1. Clinical Patient Characteristics

Between February 2020 and March 2021, a positive SARS-CoV-2 infection was demonstrated by PCR in eighteen LTRs. The median age was 49.9 years, and most patients were male (n = 12, 67%). The demographic and clinical characteristics are summarized in Table 1. The mean time since transplantation was 5.5 years, with cystic fibrosis and COPD being the most common pretransplant underlying disease (n = 6, 33% and n = 5, 28%, respectively).

**Table 1.** Patient characteristics according to COVID-19 disease severity.

|  | Mild (Siddiqi I) n = 7 | Moderate (Siddiqi IIA,B) n = 9 | Severe (Siddiqi III) n = 2 |
|---|---|---|---|
| Age, mean yrs., (range) | 41.3 (19–64) | 54.7 (28–68) | 58.5 (56–61) |
| Male sex (%) | 3 (43%) | 7 (78%) | 2 (100%) |
| BMI, mean (kg/m$^2$) | 22.4 | 26.3 | 31.2 |
| Pretransplant diagnosis |  |  |  |
| Cystic fibrosis | 4 (57%) | 2 (22%) | 0 |
| COPD | 1 (14%) | 4 (44%) | 0 |
| ILD | 2 (29%) | 2 (22%) | 2 (100%) |
| PAH | 0 | 1 (11%) | 0 |
| Comorbidities (%) |  |  |  |
| Hypertension | 2 (29%) | 6 (56%) | 2 (100%) |
| Chronic kidney disease | 2 (29%) | 4 (44%) | 2 (100%) |
| Diabetes | 2 (29%) | 3 (33%) | 1 (50%) |
| Cardiovascular disease | 1 (14%) | 3 (33%) | 2 (100%) |
| Malignancy | 0 | 4 (44%) | 1 (50%) |

BMI = body mass index.

The most common symptoms were fever (n = 9, 50%), nausea/vomiting (n = 7, 39%), cough (n = 6, 33%), dyspnea (n = 6, 33%) and fatigue (n = 6, 33%). Headache was reported by five patients (28%), and anorexia by three patients (23%). Only one patient (6%) reported altered sense of smell and taste (Table 2). Among the comorbidities, 10 patients (56%) had hypertension, six (33%) diabetes mellitus, six (33%) cardiovascular disease, eight (44%) chronic kidney disease and five (28%) a history of malignancy. Among the immunosuppressive drugs, 18 patients (100%) were long-term treated with prednisone, 15 patients (83%) had mycophenolate mofetil (MMF), 10 patients (56%) were on tacrolimus treatment, six patients (33%) had cyclosporine A, one patient (6%) had everolimus and one (6%) had rapamycin. Severity stages were classified according to Siddiqi, ranging from stage I to III. Seven (39%) patients had mild disease (stage I), five (28%) had moderate disease without hypoxemia (stage IIA), four (22%) with hypoxemia (stage IIB), while two (11%) were categorized as severe (stage III).

### 3.2. Laboratory and Radiological Features

The most notable laboratory findings were elevated levels of CRP, ferritin and D-dimers.

Computed Tomography (CT) of the chest was performed in all hospitalized patients, and showed ground-glass opacities (GGO) in 11 out of 11 (100%) patients, of whom nine (82%) had GGO combined with pulmonary consolidations. In two patients (18%), there was also a small pleural effusion, in which a thorax drainage was not indicated. One patient (9%) had a pneumothorax, which required insertion of a chest tube.

### 3.3. Therapeutic Intervention

Treatment data are mentioned in Tables 3 and 4: Six (33%) patients received remdesivir, five (28%) intravenous dexamethasone either alone or in combination with remdesivir, and 15 (83%) were treated with broad-spectrum antibiotics including co-amoxicillin, tazobactam-piperacillin, and meropenem. One patient was treated with COVID-19 convalescent plasma (CCP). Four (22%) patients were transferred to the intensive care unit, and two patients required invasive mechanical ventilation who could not be successfully extubated and died.

**Table 2.** Symptoms, signs and laboratory values.

| | Mild (Siddiqi I)<br>n = 7 | Moderate (Siddiqi IIA,B)<br>n = 9 | Severe (Siddiqi III)<br>n = 2 |
|---|---|---|---|
| Symptoms | | | |
| Fever | 2 (29%) | 7 (78%) | 0 |
| Cough | 1 (14%) | 5 (56%) | 0 |
| Dyspnea | 0 | 4 (44%) | 2 (100%) |
| Sore throat | 0 | 0 | 0 |
| Fatigue | 1 (14%) | 4 (44%) | 1 (50%) |
| Anorexia | 1 (14%) | 2 (22%) | 0 |
| Diarrhea | 1 (14%) | 2 (22%) | 0 |
| Nausea/vomiting | 3 (43%) | 3 (33%) | 1 (50%) |
| Altered sense of smell | 0 | 0 | 0 |
| Altered sense of taste | 0 | 1 (11%) | 0 |
| Headache | 2 (29%) | 3 (33%) | 0 |
| Rhinorrhea | 1 (14%) | 1 (11%) | 0 |
| Vital signs | | | |
| Temperature (°C) | 38.4 (37.9–38.8) | 37.3 (35.9–39.2) | 38.3 (36.8–39.8) |
| Heart rate (bpm) | 118 | 82 (69–129) | 88 (81–95) |
| Oxygen saturation, % | 96 | 96.4 (95–99) | 92 (90–93) |
| Laboratory values, mean (range) | | | |
| CRP (mg/l) | 29.8 (4–77) | 58.2 (4.8–140) | 302 (199–406) |
| Hemoglobin (g/l) | 135 (104–172) | 121 (96–152) | 113 (99–127) |
| Thrombocytes (G/l) | 201 (153–313) | 186 (126–365) | 174 (172–176) |
| Leucocytes (G/l) | 6.41 (3.67–10.6) | 6.67 (4.99–9.38) | 11.6 (5.9–17.3) |
| Neutrophils (G/l) | 4.89 (3.26–8.03) | 5.29 (2.04–7.49) | 10.4 (4.1–16.1) |
| Eosinophils (G/l) | 0.067 (0–0.14) | 0.024 (0–0.08) | 0 (0–0) |
| Lymphocytes (G/l) | 0.95 (0.17–1.25) | 0.60 (0.18–1.58) | 1.29 (0.47–2.1) |
| ASAT (U/l) | 42.7 (18–69) | 29.9 (14–44) | 44 (39–49) |
| ALAT (U/l) | 47.3 (11–90) | 24.7 (12–39) | 16 (9–23) |
| LDH (U/l) | 383 (309–450) | 486 (5–636) | 801 (500–1101) |
| Bilirubin (µmol/l) | 17.7 (5–34) | 8.1 (4–15) | N/A |
| Creatinine (µmol/l) | 119 (90–166) | 231 (17–809) | 208 (202–213) |
| Creatinin kinase (U/l) | 46 (43–49) | 113 (25–525) | 92 (69–114) |
| Blood Group (A, B, AB, 0) | | | |
| A (%) | 3 (43%) | 7 (78%) | 0 |
| B (%) | 0 | 1 (11%) | 0 |
| AB (%) | 0 | 0 | 0 |
| 0 (%) | 4 (57%) | 1 (11%) | 2 (100%) |
| AIFELL Score at presentation | 1.5 (1–2) | 3.2 (2–4) | 4.5 (4–5) |

**Table 3.** Treatment strategies.

| | Mild (Siddiqi I) n = 7 | Moderate (Siddiqi IIA,B) n = 9 | Severe (Siddiqi III) n = 2 |
|---|---|---|---|
| Immunosuppression | | | |
| Prednisone | 7 (100%) | 9 (100%) | 2 (100%) |
| Mycophenolate mofetil | 6 (86%) | 7 (78%) | 2 (100%) |
| Cyclosporine A | 1 (14%) | 3 (33%) | 2 (100%) |
| Tacrolimus | 6 (86%) | 4 (44%) | 0 |
| Everolimus | 0 | 1 (11%) | 0 |
| Rapamycin | 1 (14%) | 0 | 0 |
| Treatment | | | |
| Remdesivir | 0 | 6 (67%) | 0 |
| Augmentin | 3 (43%) | 0 | 1 (50%) |
| Ceftriaxone | 0 | 0 | 0 |
| Tazobactam/piperacillin | 1 (14%) | 3 (33%) | 1 (50%) |
| Meropenem | 0 | 3 (33%) | 1 (50%) |
| Vancomycin | 0 | 1 (11%) | 0 |
| Dexamethasone | 0 | 3 (33%) | 2 (100%) |
| Treatment setting | | | |
| Ambulant | 6 (86%) | 0 | 0 |
| Hospital, normal ward | 1 (14%) | 7 (78%) | 0 |
| Hospital, intermediate care | 0 | 0 | 0 |
| Hospital, intensive care | 0 | 2 (22%) | 2 (100%) |
| Hospitalization (days) | 7 (7–7) | 22 (3–44) | 20 (19–20) |
| Oxygenation | | | |
| Normal, room air | 7 (100%) | 5 (56%) | 0 |
| Oxygen, nasal cannula | 0 | 3 (33%) | |
| Oxygen, non-rebreather | 0 | 0 | 0 |
| Oxygen, HFOT | 0 | 1 (11%) | 0 |
| Non-invasive ventilation | 0 | 0 | 0 |
| Mechanical ventilation | 0 | 0 | 2 (100%) |
| ECMO | 0 | 0 | 0 |
| Outcome | | | |
| Alive | 7 (100%) | 9 (100%) | 0 |
| Dead | 0 | 0 | 2 (100%) |

ECMO = extracorporeal membrane oxygenation; HFOT = high-flow oxygen therapy.

**Table 4.** Patient characteristics, treatments and outcome.

| Patient Number | 1 | 2 | 3 | 4 | 5 | 6 | 7 | 8 | 9 | 10 | 11 | 12 | 13 | 14 | 15 | 16 | 17 | 18 |
|---|---|---|---|---|---|---|---|---|---|---|---|---|---|---|---|---|---|---|
| Date of COVID-19 | 10/2020 | 10/2020 | 03/2020 | 10/2020 | 10/2020 | 11/2020 | 11/2020 | 11/2020 | 12/2020 | 05/2020 | 11/2020 | 12/2020 | 01/2021 | 12/2020 | 01/2021 | 01/2021 | 01/2021 | 01/2021 |
| Demographic | | | | | | | | | | | | | | | | | | |
| Age (years) | 56 | 28 | 56 | 48 | 38 | 22 | 68 | 66 | 27 | 64 | 49 | 68 | 64 | 34 | 19 | 61 | 67 | 63 |
| male/female | m | m | f | m | m | f | m | m | m | m | f | m | f | m | f | m | f | m |
| BMI, kg/m2 | 31 | 27.4 | 21.9 | 19.3 | 19.6 | 20.3 | 42.8 | 27.1 | 19.5 | 27.7 | 18.9 | 22.4 | 27.4 | 14.9 | 15.2 | 31.4 | 37.5 | 31.8 |
| Transplant Data | | | | | | | | | | | | | | | | | | |

**Table 4.** *Cont.*

| Patient Number | 1 | 2 | 3 | 4 | 5 | 6 | 7 | 8 | 9 | 10 | 11 | 12 | 13 | 14 | 15 | 16 | 17 | 18 |
|---|---|---|---|---|---|---|---|---|---|---|---|---|---|---|---|---|---|---|
| Transplant year | 2019 | 2019 | 2019 | 2019 | 2006 | 2019 | 2015 | 2020 | 2014 | 2018 | 2016 | 2012 | 2016 | 2016 | 2020 | 2016 | 2010 | 2018 |
| Previous disease | ILD | PAH | COPD | CF | CF | CF | COPD | IPF | CF | IPF | Pl.par.Fi | COPD | COPD | CF | CF | ILD | ILD | COPD |
| **Comorbidities** | | | | | | | | | | | | | | | | | | |
| Hypertension | 1 | 1 | | | 1 | | | | 1 | 1 | 1 | 1 | 1 | | | 1 | | 1 |
| Diabetes | | | | | 1 | | 1 | 1 | | | | | | 1 | | 1 | 1 | |
| Cardiovascular disease | 1 | | | | | | | | | | | 1 | 1 | | | 1 | 1 | 1 |
| Malignancy | 1 | | | | 1 | | 1 | 1 | | | 1 | | | | | | | |
| Chronic kidney disease | 1 | | | | 1 | | | | 1 | 1 | 1 | 1 | 1 | | | 1 | | |
| **Immunosuppression** | | | | | | | | | | | | | | | | | | |
| Prednison | 1 | 1 | 1 | 1 | 1 | 1 | 1 | 1 | 1 | 1 | 1 | 1 | 1 | 1 | 1 | 1 | 1 | 1 |
| MMF | 1 | 1 | 1 | 1 | | 1 | 1 | | | 1 | 1 | 1 | 1 | 1 | 1 | 1 | 1 | 1 |
| Ciclosporine | 1 | | 1 | | | | | | | | | 1 | | | | 1 | 1 | 1 |
| Tacrolimus | | 1 | 1 | | | 1 | 1 | | 1 | 1 | 1 | | 1 | 1 | 1 | | | |
| Everolimus | | | | | 1 | | | | | | | | | | | | | |
| Certican/rapamy | | | | | | | | | 1 | | | | | | | | | |
| **Risk stratification** | | | | | | | | | | | | | | | | | | |
| Siddiqi Stage | III | IIA | I | IIB | IIB | I | IIA | IIA | I | I | IIA | IIB | IIB | I | I | III | I | IIA |
| AIFELL Score | 4 | 1 | 2 | 4 | 3 | | 3 | 4 | | | 2 | 4 | 4 | 1 | | 5 | | |
| | | | | | | | | | | | | | | | | | | |
| **Radiology** | | | | | | | | | | | | | | | | | | |
| Examination (CT/CXR) | CT | CT | CT | CT | CT | | CT | CT | | | CT | CT | CT | | | CT | | CXR |
| GGO | 1 | 1 | 1 | 1 | 1 | | 1 | 1 | | | 1 | 1 | 1 | | | 1 | | |
| Consolidation | 1 | | | 1 | 1 | | | 1 | | | 1 | 1 | 1 | | | 1 | | 1 |
| Pleural Effusion | | | | 1 | | | 1 | | | | | | | | | | | |
| Pneumothorax | 1 | | | | | | | | | | | | | | | | | |
| **Treatment** | | | | | | | | | | | | | | | | | | |
| **Antiviral** | | | | | | | | | | | | | | | | | | |
| Remdesivir | | | | 1 | 1 HD dose | | | 1 | | | | 1 HD dose | 1 | | | | | 1 |
| **Antibiotics** | | | | | | | | | | | | | | | | | | |
| Azithromycin | | | | | | | | | | | | | | | | | | |
| Augmentin | | | | | | | | | 1 | | | | | 1 | | 1 | 1 | |
| Ceftriaxon | | 1 | | | | | | | | | | | | | | | | 1 |
| Tazobac | 1 | | 1 | | | | | 1 | | | 1 | 1 | | | | | | |
| Meropenem | 1 | | | | 1 | | 1 | | | | | | 1 | | | | | |
| Vancomycin | | | | | | | | | | | | | 1 | | | | | |
| **Corticosteroids** | | | | | | | | | | | | | | | | | | |
| Prednison | 1 | | | | | | | | | | | | | | | | | |
| Dexamethasone | 1 | | | | | | | 1 | | | | | 1 | | | 1 | | 1 |
| CCP | | | | | | | | | | | | | | | | | 1 | |
| **Treatment Setting** | | | | | | | | | | | | | | | | | | |
| Ambulant | | | | | | 1 | | | 1 | 1 | | | | 1 | 1 | | 1 | |
| Hosp. normal ward | | 1 | 1 | | 1 | | 1 | 1 | | | 1 | 1 | | | | | | 1 |
| Hosp. ICU | 1 | | | 1 | | | | | | | | | 1 | | | 1 | | |
| No. of Hosp. Days | 20 | 3 | 7 | 43 | 44 | | 9 | 26 | | | 3 | 36 | 7 | | | 19 | | 23 |
| **Oxygenation** | | | | | | | | | | | | | | | | | | |
| Normal breathing no O2 | | 1 | 1 | | | 1 | 1 | 1 | 1 | 1 | 1 | | | 1 | 1 | | 1 | 1 |
| Oxygen nasal cannula | | | | | 1 | | | | | | | 1 | 1 | | | | | |
| Oxygen HFOT | | | | 1 | | | | | | | | | | | | | | |
| Mechanical ventilation | 1 | | | | | | | | | | | | | | | 1 | | |
| **Outcome** | | | | | | | | | | | | | | | | | | |
| Alive | | 1 | 1 | 1 | 1 | 1 | 1 | 1 | 1 | 1 | 1 | 1 | 1 | 1 | 1 | | 1 | 1 |
| Dead | 1 | | | | | | | | | | | | | | | 1 | | |

BMI = body mass index; CCP = COVID convalescent plasma; CT = computed tomography of the chest; CXR = chest x-ray; GGO = ground-glass opacities; HD dose = hemodialysis dose; HFOT = high flow oxygen therapy; Hosp. = hospitalization; ICU = intensive care unit; ILD = interstitial lung disease; IV = intravenous; MMF = mycophenolate mofetil; No. = number; PAH = pulmonary arterial hypertension; Pl.par Fib = pleuroparenchymal fibroelastosis.

### 3.4. Remdesivir in Patients with Impaired Renal Function

Remdesivir, a therapy originally developed to treat hepatitis C and Ebola, is used in treating selected patients with COVID-19. In three (17%) patients, due to an impaired renal function, remdesivir formally was contraindicated. One of them was on chronic intermittent hemodialysis, the other two had an eGFR < 30 mL/min. In patients with hemodialysis (HD), to the best of our knowledge, there are no clinical data or guidelines how to treat these patients with remdesivir. At the University Hospital of Zurich, remdesivir in HD patients, starts with 200 mg on the first day of HD, followed by 100 mg on the third and fifth day.

On the first day, the maximal concentration (Cmax) is measured immediately after the first remdesivir infusion, followed by measurements of Cmax after 2–3 h. On the second day, the minimal concentration, Cmin, is measured 24h after the first remdesivir dose. On day three, Cmin is measured before and after dialysis, and Cmax immediately after the second dose and also three hours later. On the fourth day, Cmin is measured 24h after the second dose (trough). On the fifth day, Cmin is measured before HD and after HD. The Cmax is measured after the infusion of the last dose, as well as 3 h later. Currently, the Cmax values are experimentally and help to find the area under the curve (AUC), after which dose adjustments can be considered. However, in case the dose is not in the target range, dosage adjudgments are empirical. Additionally, in patients with renal failure but without HD, remdesivir treatment consists of three doses: On the first day 200 mg are given followed by remdesivir 100 mg every 48 h (remdesivir on day 1, 3 and 5).

### 3.5. Autopsy in a Patient with COVID-19 with Siddiqi Stage III Disease

One 56-year-old patient with a bilateral lung transplantation 2 years ago due to interstitial lung disease, probably IPF, died 16 days after being diagnosed with COVID-19 by detection of SARS-CoV-2 infection. The cause of death was a severe ARDS. He presented in the emergency department of our hospital, reporting fever, acute dyspnea and a non-productive cough. On admission, he had a temperature of 39.5 °C (auricular), a heart rate of 95/min., an oxygen saturation of 93% without supplemental oxygen, a blood pressure of 146/95 mmHg and a normal heart- and lung auscultation. Among the relevant comorbidities, the patient suffered from an impaired renal function due to calcineurin inhibitors, worsened by the acute viral infection (at admission eGFR 16 ml/min., previously 30 mL/min.), and had been diagnosed with central and subsegmental pulmonary embolism four months before admission. He was on coumarin treatment for this. CT of the chest showed a right-sided consolidation in the upper lobe, with concomitant right-sided GGO, without signs of air trapping. After four days, the chest CT showed severe progression with new bilateral infiltrates in the lower lobes, and progression of GGO and bilateral crazy-paving pattern). Due to severe ARDS the patient needed intubation for invasive mechanical ventilation. He was treated with dexamethasone 6 mg iv. for 10 days (after 10 days, prednisone 15 mg was continued), remdesivir and pragmatically with meropenem, levofloxacin, and amphotericin. The immunosuppression with ciclosporin was continued; however, MMF was discontinued due to lymphocytopenia and thrombocytopenia. Six days later, still on mechanical ventilation support, he developed an obstructive shock due to a total left-sided tension pneumothorax, probably due to barotrauma (Figure 1a–e). Initially, he was treated with chest tube drainage (20 Ch.), sand after two days a second chest tube (28 Ch.) was inserted, followed by another chest tube (28 Ch.) two days later due to insufficient lung expansion. Other complications in this patient were atrial fibrillation, protracted thrombocytopenia and renal failure requiring hemofiltration.

In the bronchoalveolar lavage (BAL), performed immediately after intubation, PCR of SARS-CoV-2 was positive. In the BAL, no microorganisms were grown, and the Aspergillus antigen (galactomannan) was negative (index < 0.5). Bronchoscopy showed no endobronchial mucus retention (Figure 1f,g). In addition, the serological Aspergillus antigen was negative (index < 0.5), so was the PCR for Bordetella pertussis and parapertussis. Chlamydia pneumoniae and psittaci, Legionella pneumophila and other species, as well as

Mycoplasma pneumoniae were all negative in the BAL. The aerobic and anaerobic samples from blood cultures were negative. Microbiological investigations of the vena jugularis and arteria radialis catheters were negative. The patient still had considerable subcutaneous emphysema (Figure 1h) developed progressive multiorgan failure and after 16 days in the intensive care unit, treatment was discontinued after the unanimous decision of the medical team and the family of the patient.

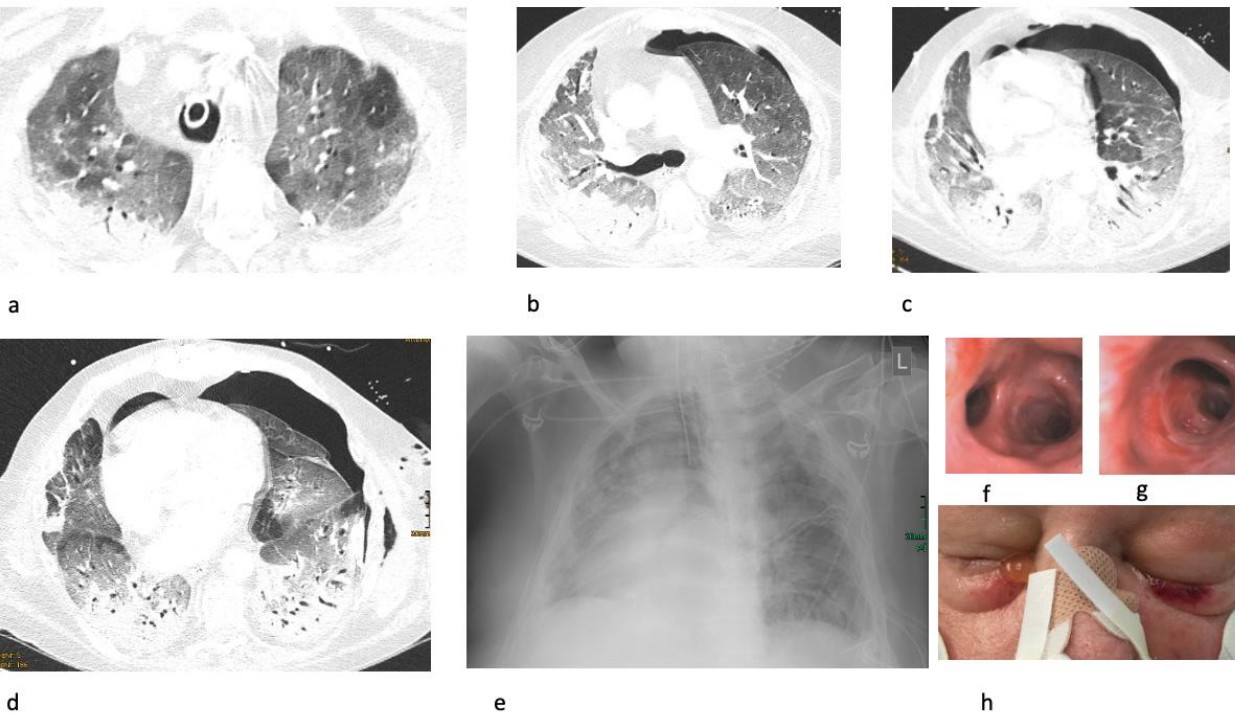

**Figure 1.** CT scan, chest X-ray, bronchoscopy images, and subcutaneous emphysema. CT scan showing ARDS, GGO and pneumothorax (**a–d**); chest X-ray after pneumothorax drainage (**e**); bronchoscopy without endobronchial mucus retention (**f**,**g**); massive subcutaneous emphysema (**h**).

The patient died that same day, and an autopsy was performed. The post mortal SARS-CoV-2 PCR was still positive, and the SARS-CoV-2 concentration in the tracheo-bronchial secretion showed 56 million SARS-CoV-2 copies/mL. Pathology examination further showed the typical COVID-19 diffuse endothelitis in both lungs and in the epi-cardial and intramyocardial blood vessels. Capillaries in heart and lung showed diffuse peripheral thrombi with fibrine and leucocytes, which are also well-known COVID-19 findings. In the lung parenchyma, there was extensive multifocal hemorrhagic infarction, and multifocal acute bronchopneumonia, as well as diffuse alveolar damage (DAD). The multifocal acute bronchopneumonia suggested bacterial superinfection, although (with ongoing broad-spectrum antibiotics and amphotericin B) no bacteria or Aspergillus could be cultured in any of the samples. An additional finding in the autopsy was an early stage bowel necrosis and a centrilobular hepatic necrosis.

## 4. Discussion

In this case series, 18 LTRs with different severity degrees of COVID-19 have been described. Although we have a small number of patients and statistics are descriptive, there tends to be a male predominance with elderly patients with a more severe COVID-19 stage (IIB and III). The extrapulmonary symptoms were predominant in most of the LTRs in our case series.

As has been seen in immunocompetent patients as well, in our case series the LTRs had a more severe COVID-19 stage with a higher body mass index (BMI). The mortality in severe COVID-19 (stage III) was 100%, and these patients were not only older and had

higher BMI values, but they also all had the other COVID-associated main risk factors, namely hypertension, chronic kidney disease and cardiovascular disease. Moreover, the patients with severe COVID-19 also had a higher CRP value, lower hemoglobin, higher leucocytes, neutrophils, and higher LDH values. In contrast to the literature reports on immunocompetent patients with COVID-19, the transaminases in our patients, did not show relevant elevations.

Most patients had blood group A, followed by 0. Both patients with severe COVID-19 had blood group 0.

The AIFELL score, a triage tool used to assess risk in COVID-19 patients, was also low in the Siddiqi stage I, higher in stage II and the highest in stage III patients. All patients were treated with long-term immunosuppressive drugs in the pre-COVID stage, including prednisone in all patients, and in most patients MMF and a calcineurin inhibitor. The most common calcineurin inhibitor was tacrolimus. Six patients (two of them had chronic hemodialysis) have been treated with remdesivir, and five with dexamethasone. One patient was treated with COVID-19 convalescent plasma.

Only 6 out of 18 patients were treated in the ambulatory setting, all other patients were hospitalized, with a mean hospitalization duration of 20 days.

In the following paragraphs, we will discuss some of the above-mentioned special aspects of our case series.

### 4.1. COVID-19 Severity: The Siddiqi Stages

COVID-19 in most patients in our case series was classified as mild or moderate, according to the Siddiqi classification. The Siddiqi stages classify COVID-19 disease states and potential therapeutic targets [5]. This classification has three escalating phases. Stage I (early infection), characterized by a viral response phase, with clinical symptoms including mild constitutional symptoms, fever, a dry cough, and laboratory shows lymphopenia. In this phase, immunosuppression should be reduced, and excess systemic steroids be avoided. In Stage II (pulmonary phase), there is both a viral response phase and a host inflammatory response phase. Clinical symptoms include dyspnea without hypoxia (IIA) and with hypoxia (IIB), and chest imaging is abnormal (infiltrates), laboratory values showing elevated transaminases, and procalcitonin is generally low or normal. In this stage, mycophenolate should be reduced, according to the ISHLT guidance document [7]. In Stage III (hyperinflammation phase), there is solely a host inflammatory response, with acute respiratory distress syndrome (ARDS), systemic inflammatory response syndrome (SIRS) and cardiac failure, and laboratory values show elevated inflammatory markers (CRP, LDH, Interleukin-6, D-dimer, ferritin), troponin, and N-terminal pro-b-type natriuretic peptide (NT-proBNP). In Stage III, mycophenolate should also be discontinued [7]. Patients in Stage II and III are likely to be the patients that would benefit most of the continued use of calcineurin inhibitors, attenuating the hyperinflammation (cytokine storm).

### 4.2. SARS-CoV-2 Infection: Extrapulmonary Manifestations

Although infection in the respiratory system is the most important manifestation of COVID-19, there are also important extrapulmonary disease manifestations. In our case series, the extrapulmonary manifestations were the predominant symptoms. Extrapulmonary COVID disease can result in gastrointestinal, cardiovascular, renal and neurological morbidity. Pathophysiologically, there is severe microvascular thrombosis and inflammation, resulting from vascular endothelial dysfunction [8,9]. Interestingly, endothelial dysfunction is also present in the high-risk population for severe COVID-19, in particular patients with hypertension, obesity and diabetes [8]. These comorbidities were also relevant in the patients in our case series, especially in the two patients with severe COVID-19 who did not survive. The patient in which autopsy was performed also showed extensive endothelial dysfunction, as has been described above.

In the literature, most COVID-19 fatalities in immunocompetent patients were observed in elderly men with the aforementioned comorbidities. Although the exact patho-

physiology of endothelial dysfunction in COVID-19 needs further clarification, the scientific evidence suggests that COVID-19 targets endothelial cells [10].

### 4.3. SARS-CoV-2 Infection: Pulmonary Manifestations

The autopsy report on our patient showed an important pulmonary component of COVID-19. Inflammation is a key component of SARS-CoV-2 infection. Research has shown that there are similarities between the acute respiratory distress syndrome (ARDS) and the acute radiation syndrome (ARS) [11]. The ARS, being a public health emergency, occurs after exposure to high doses of radiation and leads, as in ARDS, to a cytokine storm, with remarkably similar pathophysiology, including increased pro-inflammatory molecules and decreased other anti-inflammatory molecules [11]. Medical treatment strategies for ARS could be helpful in the clarification of the COVID-19 induced ARDS [11]. Among the medical countermeasures in ARS are growth factors, antioxidants, anti-inflammatory agents, anti-fibrotic drugs, RAS-targeted approaches, and treatment for vascular injury such as statins. These approaches are potentially of interest since treatment options for COVID-19 are still very limited and there is no cure for COVID-19. If all available treatment strategies fail to prevent post-COVID-19 lung fibrosis, lung transplantation may be the only curative option in selected severe cases without signs of improvement over weeks or months. Patients recovering from severe COVID-19, especially after ARDS, have a high risk of developing pulmonary fibrosis [12].

### 4.4. Liver Function Test Abnormalities

In our case series, the liver function (both transaminases and bilirubin levels) at presentation and during the course of the disease was unremarkable, even in the severe cases. This is in contrast with the literature on immunocompetent patients with COVID-19, in whom liver function test abnormalities are associated with a severe course of the SARS-CoV-2 infection [13]. In a meta-analysis, including 3428 patients from 20 retrospective studies, liver dysfunction was significantly higher in critically ill patients with unfavorable outcomes in COVID-19 [14,15], as could also be observed in other coronaviral diseases (SARS and MERS) [16–18]. Studies show the incidence of liver injury ranging from 58–78%, presenting with elevated transaminases and bilirubin levels [19,20]. Autopsy studies show mild lobular and portal activity along with microvascular stenosis [14,21–23]. In case of hepatic involvement in COVID-19, this can be a direct cytopathic effect such as in hyper-inflammation (cytokine storm) and sepsis, or a drug-induced liver injury. Interestingly, cholangiocytes have a higher ACE2 receptor expression, which makes the liver a potential target for SARS-CoV-2 [15]. In the literature, a higher proportion of liver enzyme elevation was observed in patients receiving lopinavir/ritonavir treatment (56.1% vs. 25%) [24], but in our case series none of the LTR received this drug treatment. However, one study in patients with COVID-19 showed liver injury in 10–13% of patients treated with remdesivir [25]. We did not observe this in our cohort (data on evolution of liver enzymes not shown).

### 4.5. CRP as Marker of Disease Activity in COVID-19

In our cases the patients with the highest CRP values within 3 days of disease onset were those that had the most severe COVID-19 disease evolution. All patients with a CRP level ≥199 mg/L were treated in the ICU and died. CRP is a marker of COVID-19 disease activity. It is a plasma protein that is produced in the liver. Various mediators of inflammation, such as Interleukin (IL-)6, can induce CPR production. Elevation of CRP levels in COVID-19 are associated with the severity of COVID-19. Compared to the erythrocyte sedimentation rate (ESR), CRP levels were significantly greater during early periods of severe COVID-19 cases and were shown to be a more sensitive biomarker in reflecting disease development [26,27]. In a retrospective study, the majority of patients with severe COVID-19 showed significantly higher CRP levels as in the non-severe COVID-19 patients (57.9 mg/L vs. 33.2 mg/L, $p < 0.001$) [28]. Another retrospective study showed

that CRP can effectively assess disease severity and predict outcomes in patients with COVID-19, with an increased risk of progression to a higher severity stage in patients with CRP levels >41.8 mg/L [29–31]. CRP values were found to be a more reliable indicator for earlier identification of case severity than CT scans alone [26].

### 4.6. COVID-19 and ABO Blood Group

Here, we could show no clear influence of ABO blood group on outcome. The two fatal cases had blood group 0. Several studies showed an association between ABO blood groups and the risk of SARS-CoV-2 pneumonia [32].

Data from Wuhan (medRxiv, preprint, not peer-reviewed) including 2173 COVID-19 patients, showed that blood group A was overrepresented in COVID-19 compared with non-A blood groups. In contrast, patients with blood group O showed a significantly lower risk for the infection compared with non-O blood groups [33].

Similar results were shown by the Presbyterian hospital in New York, including 1559 patients with COVID-19 [34].

Gérard et al. studied the ABO blood group in patients with COVID-19, by comparing the patients (n = 1888) possessing anti-A in their serum (i.e., those of B and O blood groups) and those who did not (i.e., those of A and AB blood groups) to the control cohort (n = 3694) [35]. They found significantly less COVID-19 in patients with anti-A in serum (i.e., B and O blood groups) compared to those lacking anti-A (i.e., A and AB blood groups), showing a possible protective effect of anti-A. Surprisingly, Gérard et al. also found a difference between anti-A from O and anti-A from B. The anti-A from O showed an underrepresentation in COVID-19 and anti-A from B an overrepresentation, indicating that anti-A from O is more protective than anti-A from B [35]. This important difference could be related to the isotype of antibodies, being anti-A isotype IgM in serum from blood group B patients, but IgGs in blood group O serum. Although several studies have shown a higher risk of SARS-CoV-2 infection for certain blood groups, one meta-analysis has shown that there was no clear correlation between blood groups and the severity of COVID-19 [36].

### 4.7. AIFELL Score

The AIFELL score, developed by Levenfus et al., is a clinical prediction score used for triage purposes to assess risk and provide guidance of further diagnostic and therapeutic steps in patients suspected to have COVID-19 [6]. It can be easily applied in emergency wards, and also has an interactive website (www.aifell.net) correlating the AIFELL score to the above-mentioned Siddiqi stages. In this way, it can be used to select probable COVID-19 cases for hospitalization. Unfortunately, this score has not been evaluated in LTRs yet, but seems promising in this group of patients as well. Altered smell and/or taste is one aspect assessed by the AIFELL score, and this disease feature was only noted in one patient in this case series. Although this symptom appears to be fairly frequent in the general population, it was hardly noted in LTRs.

### 4.8. Remdesivir in Patients with Impaired Renal Function

In this case series, six (33%) patients were treated with remdesivir, of these three had kidney failure (two were on chronic intermittent hemodialysis), which is generally considered a contraindication for remdesivir treatment.

Remdesivir (GS-5734) is a nucleoside analog with a broad antiviral activity to RNA viruses and is also still under investigation for the treatment of Ebolavirus (EBOV, the primary indication), MERS and SARS-CoV-1 [37]. As the first approved drug for COVID-19, remdesivir remains somewhat controversial in the treatment of COVID-19. Both the SOLIDARITY trial [38] and the ACTT-1 trial [39] did not show a significant survival benefit. Nevertheless, in the ACTT-1 trial, remdesivir was shown to be superior to placebo in shortening the time to recovery in adults who were hospitalized with COVID-19, and showed evidence of lower respiratory tract infection [39]. In the ACTT-1 trial, the median

recovery time improved from 15 to 10 days in patients with remdesivir compared to placebo, respectively. The lower 14-day mortality rate in the remdesivir treatment group may indicate a beneficial effect, although it was not statistically significant [39]. However, that study was not powered to evaluate mortality, and therefore mortality should be further evaluated in larger studies with different stages of COVID-19. Reducing the time to recovery by 31%, remdesivir may therefore help to reduce the number of inpatient days, with potential positive effects on hospital costs and capacity issues during the pandemic [37]. Taking the results of both trials together, it might be concluded that treating patients "relatively late" in the course of the disease, remdesivir will not improve the mortality rate in patients with COVID-19. Defining the right timing for the use of remdesivir in COVID-19 still needs further studies. Both trials had a very heterogeneous study population, making firm conclusions on remdesivir treatment difficult. If effective at all, remdesivir should probably be given early in the disease process. Currently, studies with remdesivir treatment in COVID-19 accounting for disease severity and measuring viral load of SARS-CoV-2 are still awaited. Patients with high viral loads are possibly the best candidates to treat with the antiviral drug remdesivir, but this still needs to be proven.

An important issue, especially relevant in the LTR population, is the question of how to treat patients with an impaired renal function. As a prodrug, remdesivir is predominantly metabolized by hepatic enzymes with hydrolase activity [40,41]. Routine monitoring of liver function tests is recommended, and remdesivir should be discontinued in patients with alanine aminotransferase (ALAT) $\geq$10 times the upper limit of normal. The proposed standard dosage is 200 mg as a single dose on day 1, followed by 100 mg once daily. In patients with an eGFR $\leq$30 mL/min and in patients with renal replacement therapies, remdesivir is not recommended.

However, a multicenter, retrospective study reviewing hospitalized patients with SARS-CoV-2 who received remdesivir, showed that this remdesivir treatment was not significantly associated with increased acute kidney injury (AKI) at the end of treatment in patients with an eGFR <30 mL/min compared to patients with an eGFR $\geq$30mL/min [42]. The advice not to prescribe remdesivir in patients with an eGFR $\leq$30 mL/min can be understood in the light of the paucity of clinical data in this group of patients. In animal experiments, remdesivir has been shown to be nephrotoxic in monkeys and rats; however, these doses were 2.1–3.5 fold higher than the doses used in the treatment of SARS-CoV-2 for humans [42]. Nevertheless, in the study of Ackley et al., there was a significantly higher mortality rate in patients with an eGFR <30 mL/min., attributed to the older age, more comorbidities, more frequent use of vasopressors or inotropes, and more frequent use of mechanical ventilation [42].

One case report on a LTR with an initially normal renal function who experienced renal failure after initiation of remdesivir for the treatment of COVID-19 is of concern [43]. In this patient, serum remdesivir concentrations were undetectable, but there were elevated levels of the remdesivir metabolite (GS-441524), suggesting that the metabolite could be responsible for the renal failure in this patient [42,43]. Although it should be mentioned that renal disease is a predictor of COVID-19 related mortality [44], and more than one-fourth of patients with an eGFR <30 mL/min require mechanical ventilation due to COVID-19 related respiratory failure [42], a potential renal toxicity from the accumulation of remdesivir active metabolites requires further study.

*4.9. COVID-19 Convalescent Plasma (CCP)*

One of our patients in this case series was treated with convalescent plasma collected from recovered COVID-19 patients (CCP). Currently, add-on CCP, in addition to remdesivir and dexamethasone, can be considered in patients with an immune-deficient state, such as after SOT receiving immunosuppressive therapy, in HIV/AIDS, after aplasia-inducing chemotherapy before neutrophils recovery. This is a passive immune therapy, that currently being evaluated in clinical trials. It has been shown that most individuals with laboratory-diagnosed SARS-CoV-2 infection develop not only measurable antibody responses, but

also neutralizing antibodies [45]. The neutralizing antibody levels decline within the first 3 months following diagnosis, which suggests the collection of convalescent plasma with high neutralizing antibody concentrations may be optimally performed within a short time window after the infection has resolved [45]. Studies on CCP suggest improved clinical outcomes including radiological resolution, reduction in viral loads and improved survival. Most data relating to CCP treatment comes from the non-transplant population. The study of Duan et al. showed rapidly increasing neutralizing antibodies, and significantly improved clinical symptoms along with an increase of the oxyhemoglobin saturation within 3 days. In addition, improvements of lymphocyte counts, decreased C-reactive protein and various degrees of resolution of lung lesions in the radiological examinations were observed [46].

Moreover, there was no evidence of clinical hyperimmune responses after CCP treatment in a case series with 20 critically ill patients and 20 controls [47]. In a large series of 5000 hospitalized adults with severe or life threatening COVID-19, with 66% requiring intensive care unit treatment, the transfusion of CCP showed that the mortality rate was not excessive, and suggested that transfusion of CCP is safe in hospitalized patients with COVID-19 [48]. In that series, the incidence of all serious adverse events (SAEs) in the first four hours after CCP was <1%, including a low mortality rate (0.3%) [48]. Among the SEAs were mortality (n = 4), transfusion-associated circulatory overload (TACO; n = 7), transfusion-related acute lung injury (TRALI; n = 11), and severe allergic transfusion reactions (n = 3) [48]. In an uncontrolled case series of five critically ill patients with COVID-19 and ARDS, CCP treatment showed clinical improvement, and in four patients, ARDS resolved at 12 days after CCP, and three patients were weaned from mechanical ventilation within 2 weeks of treatment [49]. Although this treatment is promising, data on LTRs with COVID-19 treated with CCP are still lacking. Our patient who received CCP survived COVID-19 and did not show clear adverse events from the CCP treatment. However, caution with CCP in LTRs is certainly advisable, since CCP in immunosuppressed patients has also been associated with the emergence of new SARS-CoV-2 variant populations in these patients. These viral mutants are more likely to arise in immunocompromised patients, as they have a higher viral burden, increasing the opportunity for variant selection [50]. Therefore, CCP use for COVID-19 in LTRs could give rise to SARS-CoV-2 mutations. This has been observed in immunosuppressed patients treated with CCP for COVID-19 [51].

This hypothesis was confirmed in animal experiments, showing that CCP resulted in antibody-resistant SARS-CoV-2 variants, including the E484K mutation associated with vaccine resistance [52]. A comparable mechanism with emergence of resistant variants has been observed in immunocompromised patients with influenza infections who received long-term oseltamivir treatment [53]. The emergence of SARS-CoV-2 variants may lead to infections in COVID-19 vaccinated LTRs or in those who have survived COVID-19.

### 4.10. Respiratory Co-Infections

In our study, 15 patients (83%) were treated with broad-spectrum antibiotics including co-amoxicillin, tazobactam-piperacillin and meropenem, in order to prevent or treat respiratory co-infections. In all 18 patients, we found no evidence of respiratory co-infections, although these infections cannot be completely ruled out.

Estimations of the prevalence of co-infections among COVID-19 patients range from 0% to 45% [54]. Most of the co-infections occur within the first 4 days after infection[55], more commonly in SOT recipients [55,56].

In SOT recipients with COVID-19, respiratory secondary co-infections have been addressed in a large number of studies, showing bacterial, viral and fungal secondary infections [57]. Bacterial secondary co-infections were due to Gram negative bacteria, including Escherichia coli, Pseudomonas aeruginosa, Acinetobacter baumanni, Klebsiella pneumoniae, Morganella morganii, and Stenotrophomonas maltophilia, as well as Gram positive bacteria, including Enterococcus faecalis, Staphylococcus aureus, and Streptococcus haemolyticus [57].

Several authors reported viral secondary co-infections due to Cytomegalovirus (CMV) infection [58]. Although a secondary co-infection with influenza virus was demonstrated in a liver transplant recipient, this has not been described in LTR yet [59]. Aspergillus fumigatus [60] and Aspergillus niger [61] are also known to cause secondary co-infection in LTR.

While bronchoscopy is not available or advisable in routine care of COVID-19 patients, it is an option in mechanically ventilated ICU patients, providing microbiological samples from broncho-alveolar lavage or bronchial wash samples. Alternatively, samples can be obtained from (blind) tracheal aspirates.

### 4.11. Immunosuppressive Therapy during COVID-19

In LTRs affected by COVID-19, the optimal management of immunosuppression still remains an unanswered question. Our patients in this case series were all on long-term immunosuppressive therapy. Immunosuppression causes lymphopenia, being a risk factor for severe COVID-19. Moreover, mycophenolate as well as mTOR inhibitors, can impair the immune response to viral (and bacterial) infections. Therefore, in COVID-19, mycophenolate often will be reduced or discontinued, as we also did during the SARS-CoV-2 infection. The risk of decreasing or pausing mycophenolate should be weighed against the risk of transplant rejection. This is in line with the current recommendation of the International Society of Heart and Lung Transplantation (ISHLT), advising to hold mycophenolate mofetil, mTOR inhibitors or azathioprine in the context of hospital admission with moderate/severe COVID-19 [7].

On the other hand, calcineurin inhibitors (CNIs) may prevent or attenuate the cytokine storm, by inhibiting interleukin (IL)-6 and IL-1 pathways, and therefore are maintained in most patients. In our 18 patients, we did not reduce or discontinue CNI.

### 4.12. Hospital Admission Rate

The clinical spectrum of COVID-19 in LTRs is broad, ranging from mild infection of the upper respiratory tract to severe acute respiratory distress syndrome with multiorgan failure and death as demonstrated here in this single center case series. The impact of maintenance immunosuppression in LTRs on COVID-19 severity, remains to be defined.

In our case series, 67% (n = 12) of patients with COVID-19 were hospitalized. One meta-analysis, studying the hospital admission rate in solid organ transplant recipients with COVID-19, showed that the hospital admission rate in these patients is significantly higher (81%) as compared to the general population [57]. However, the higher admission rates may rather reflect the defensive treatment strategy in these vulnerable patients, in whom careful clinical monitoring in a hospital setting is preferred since the respiratory deterioration in COVID-19 patients frequently is rapid and escalation of therapy is easiest in the hospital setting. In some instances, when patients fear COVID-19 deterioration, they might prefer inpatient treatment and monitoring, but sometimes also the opposite can be seen when patients fear hospitalization due to COVID-19 related overcrowding of hospitals with limited resources or they fear contracting COVID-19 in the hospital setting. The latter has been observed frequently during the early phases of the pandemic, with a general avoidance of the hospitals due to accumulation of severe cases. It should also be mentioned that studies comparing these admission rates are difficult to compare due to differences in comorbidities in LTRs compared to the general population.

### 5. Conclusions

This study reflects the experience with COVID-19 in LTRs during the first two disease waves in Switzerland and describes clinical, laboratory, radiology features and clinic outcomes. Severe disease was shown in two patients, who did not survive. The study highlights the exceptionally high rate of non-pulmonary symptoms in comparison to immunocompetent patients, and suggests that comorbidity as well as elderly and overweight patients have a higher risk of non-favorable outcomes. Laboratory values suggesting

a dismal outcome are elevated CRP and LDH, while liver functions remained normal in all stages of COVID-19 severity. In LTRs, remdesivir and CCP can be considered as treatment options depending on the disease stage. Additionally, in patients with chronic renal insufficiency, remdesivir could be considered using an adapted dosage. The rate of hospitalization in this population is relatively high, several explanations have been discussed above.

**Author Contributions:** R.H. and M.M.S.: patient treatment, concept, design, acquisition and interpretation of data, draft writing, analysis of data. S.P.: patient treatment, acquisition of data, protocol remdesivir in hemodialysis C.S., F.G., I.I.: patient treatment. M.M.S.: supervision of interpretation of study, draft writing and editing. All authors have read and agreed to the published version of the manuscript.

**Funding:** No funding was obtained for this study.

**Institutional Review Board Statement:** Swissethics, No. 2021-00293.

**Informed Consent Statement:** General informed consent available of all patients studied.

**Data Availability Statement:** Not applicable.

**Acknowledgement:** We would like to thank Nicolas Mueller for the infectious disease division for his advice given concerning the treatment of some of our patients.

**Conflicts of Interest:** The authors declare no conflict of interest.

**Limitation of Study:** The main limitation of the study is the single center design and the fairly small sample size, and thus the results may not reflect the true picture of a large cohort. The pharmacological agents used for the treatment of COVID-19 in this cohort were not proven in randomized controlled trials and were used as per institutional guidelines for management of COVID-19 patients.

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
