# Peer review of "Clinical Characteristics, Treatments and Outcomes of 18 Lung Transplant Recipients with COVID-19"

_2673-3943, doi:10.3390/transplantology2020022_

Round 1
Reviewer 1 Report
The study reports clinical characteristics of a cohort, followed at a single Institution, of 18 lung transplant patients who had COVID-19.
The authors clearly reported clinical and radiological findings from their cohort and pathological aspects obtained from one of their patients.
Major concerns:
The absence of a control group of immunocompetent COVID-19 patients is the major limitation of the study.
The discussion is over the aim of the study. It looks like a non-systematic revision of COVID-19 and is not functional to the study. It should be reduced. Moreover, the authors missed a study on covid and Ltx (PMID: 32743882).
Regarding to the pathological report of the patients, medical history of the patient should be largely reduced. This is not a case report. Please focus more on pathology aspects.
It looks like that neither radiological nor pathological aspect of COVID-19 is different in LTx from the general population. What is authors’ opinion in merit? Can the authors provide a control group?
Some CT or path imagines will certainly improve the manuscript.
Author Response
Reviewer 1 |
|
Reviewer (R) |
The absence of a control group of immunocompetent COVID-19 patients is the major limitation of the study. |
Authors (A) |
We have chosen to describe our cohort as a case series, and therefore did not include a control group. We hope this is acceptable to the reviewer. |
R |
The discussion is over the aim of the study. It looks like a non-systematic revision of COVID-19 and is not functional to the study. It should be reduced. Moreover, the authors missed a study on covid and Ltx (PMID: 32743882). |
A |
We shortened the discussion (see also comments of Reviewer 2). We thank the Reviewer for the missing reference, and included this reference in our manuscript. |
R |
Regarding to the pathological report of the patients, medical history of the patient should be largely reduced. This is not a case report. Please focus more on pathology aspects. |
A |
The format of this study is a case series (Category “case report”). As there are only very little data on lung transplant recipients with COVID-19 who underwent an autopsy, we thought it would be interesting to describe the medical history of the patient. However, we shortened the autopsy report. In order to focus more on the pathology aspects (see also the last comment of Reviewer 1) we also included CT images. |
R |
It looks like that neither radiological nor pathological aspect of COVID-19 is different in LTx from the general population. What is authors’ opinion in merit? Can the authors provide a control group? |
A |
We agree with the reviewer, in our opinion these aspects do not differ from the general population. To provide more details, we included the radiology characteristics in the new table. Regarding the control group we refer to our first response. |
R |
Some CT or path imagines will certainly improve the manuscript. |
A |
We included the CT images in our article. |

Reviewer 2 Report
This work accurately describes the outcome of 18 lung transplant patients who subsequently developed SARS-COV2 infection.
The work is interesting and comprehensively describes the clinical history of patients with the data relating to the viral infection in question with the correlation with respect to the AIFELL and Siddiqi score. The number of patients subjected to analysis is small but fairly homogeneous. The authors conclude that lung transplanted patients have a high rate of non-pulmonary symptoms and that rendesevir and CCP should be considered in high risk patients. References are adequate.
Problems:
I suggest adding a table including data at LTR , immunosuppression at covid 19 infection, given therapy and response for each single patient. This may help the reader for for a better understanding of the clinical progress of patients.
The discussion can be shortened
Author Response
R |
I suggest adding a table including data at LTR , immunosuppression at covid 19 infection, given therapy and response for each single patient. This may help the reader for for a better understanding of the clinical progress of patients. |
A |
We included the table with these data. |
R |
The discussion can be shortened |
A |
We shortened the discussion, see also Reviewer 1. |

Round 2
Reviewer 1 Report
No further comments. Thank you.
Author Response
Thank you, reviewers